# MappingFormer: Learning cross-modal feature mapping for visible-to-infrared image translation

## ABSTRACT

Due to the limitations of infrared image acquisition conditions, many essential tasks currently rely on visible images as the main source of training data. However, single-modal data makes it difficult for downstream networks to show optimal performance. Therefore, converting the more easily obtainable visible images into infrared images emerges as an effective remedy to alleviate the critical shortage of infrared data. Yet current methods typically focus solely on transferring visible images to infrared style, while overlooking the crucial infrared thermal feature during cross-modal translation. To elevate the authenticity of cross-model translation at the feature level, this paper introduces a translation network based on frequency feature mapping and dual patches contrast, MappingFormer, which can achieve cross-modal image generation from visible to infrared. Specifically, the generator incorporates two branches: low-frequency feature mapping (LFM) and high-frequency feature refinement (HFR), both have embedded the Swin Transformer blocks. The LFM branch captures the fuzzy structural from visible images, while the HFR focuses on mapping edge and texture features. The extracted dual-branch frequency features undergo refinement and fusion through cross-attention mechanisms. Additionally, a dual contrast learning mechanism based on feature patch (DFPC) is designed to infer effective mappings between unaligned cross-modal data. Numerous experimental results prove the effectiveness of this method in cross-modal feature mapping and image generation from visible to infrared. This method holds significant potential for downstream tasks where infrared data is limited.

## CCS CONCEPTS

• **Information systems** → **Multimedia content creation**; • **Computing methodologies** → *Computer vision*.

## KEYWORDS

visible-to-infrared, image translation, feature mapping, contrastive learning, cross-modal

*ACM MM, 2024, Melbourne, Australia*
© 2024 Copyright held by the owner/author(s). Publication rights licensed to ACM.
ACM ISBN 978-x-xxxx-xxxx-x/YY/MM
https://doi.org/10.1145/nnnnnnn.nnnnnnn

## 1 INTRODUCTION

In the domain of visual sensing technology, visible (VIS) image and infrared (IR) image constitute multimodal data that are frequently employed. VIS image provides rich textural details and geometric features, while IR image reveals the temperature distribution of objects and backgrounds [6, 27, 30]. However, due to the shooting limitations of infrared cameras or the absence of dependable and precise infrared simulation systems, many downstream applications struggle to collect adequate IR data to support model training, and cannot make downstream models perform well. These applications face a significant challenge of lacking IR image data [12]. Current research aimed at addressing IR data shortage mainly focuses on generating corresponding images from VIS. How to learn the mapping correlation between cross-modal data during training has become a crucial research question urgently needs to be solved [39, 41].

Currently, various solutions exist for cross-modal generation from VIS to IR images. One approach involves manually simulating the hot zone using physical models, but its efficiency and precision leave room for improvement [8, 25]. Alternatively, intelligent methods are widely used to embed the input VIS image into a latent feature space and subsequently reconstruct the corresponding IR image based on a nonlinear transformation relationship [9]. For example, using generative adversarial networks (GAN) [2, 14, 44], unsupervised learning [21, 31] and self-supervised learning [10, 28] to transform the VIS domain into an IR style. While the above transformation methods can yield outputs resembling real IR images, they still overlook the conversion of the thermal features. Recent works based on contrastive learning [5, 11, 18, 23] aim to address these limitations, but a notable semantic gap persists between the VIS and IR domains, and these methods still require additional constraints. Moreover, numerous prevalent image translation methods focus more on the diversity of translation styles[1, 17]. When directly applied to VIS-to-IR conversion task, it is difficult to generate images that conform to infrared thermal features. Noting that the data collected in real-world scenarios are often unaligned, exhibiting differences in perspective or resolution, which increases the difficulty of cross-modal conversion.

In this work, we propose a novel approach for VIS-to-IR image translation, exploring the correlations between multimodal images by mining the frequency feature, as shown in Figure 1. Our motivation stems from the observation that despite VIS and IR features have modal specificity, they inherently contain shared representations to objects or features. Meanwhile, previous studies [16, 33, 46] have emphasized that mining frequency features and reconstructing images is maybe effective for image translation. Building upon this

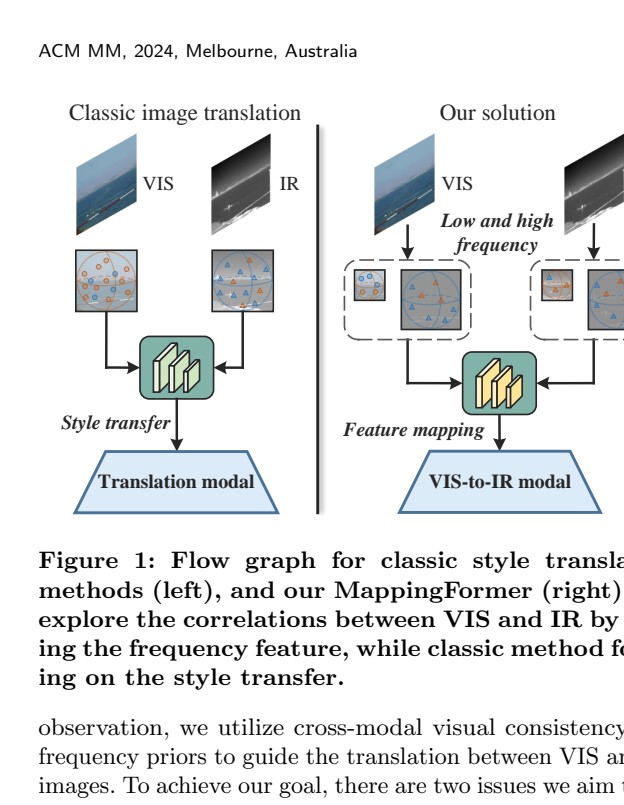

Classic image translation     Our solution

**Figure 1: Flow graph for classic style translation methods (left), and our MappingFormer (right). We explore the correlations between VIS and IR by mining the frequency feature, while classic method focusing on the style transfer.**

observation, we utilize cross-modal visual consistency and frequency priors to guide the translation between VIS and IR images. To achieve our goal, there are two issues we aim to address. The first is how to incorporate frequency features into the generative model efficiently. The self-attention and shifted window in Swin Transformer (Swin-T) [20] demonstrate remarkable perceptual ability and computational efficiency in feature extraction. Hence, integrating the Swin-T into the generator to process frequency domain information is worth investigating. Besides, the second issue is how to effectively impose constrains on cross-modal feature mapping. We postulate that incorporating the bidirectional contrastive learning into the constraints of feature mapping could enhance the quality of the asymmetric cross-modal translation.

To ensure that IR images generated from VIS accurately reflect thermal laws, we introduce a network tailored from VIS to IR, MappingFormer. It comprises two generators and two discriminators, aiming to maximize the mutual information between input and output feature patches. First, the VIS image is decomposed into low and high-frequency components in the generator. Then, dual branches are devised for low-frequency feature mapping (LFM) and high-frequency feature refinement (HFR) with embedded Swin-T. Cross-attention is employed to integrate the mapping information from both branches to generate more authentic infrared features. The LFM extracts fuzzy structural and color information, whereas HFR enhances edges and textures. Finally, by bidirectional contrast mechanism for feature patches, the mapping process of unaligned features is constrained. As such, our method diverges from general image translation methods, offering a specialized approach for cross-modal translation between VIS and IR images.

In summary, our work makes the following contributions:

- A cross-modal feature mapping network for VIS-to-IR (MappingFormer) is proposed, which ensures the translated results more related with the thermal features in

infrared. To our knowledge, this is the first work that using frequency mapping to the VIS-to-IR translation.
- Within the generator, two branches of low-frequency mapping and high-frequency refinement have been devised. And a dual contrast learning mechanism has been designed as the constrain in feature mapping process.
- The experiment results clearly support the potential of our proposed method, confirming the applicability to cross-modal translation and downstream tasks.

## 2 RELATED WORK

### 2.1 Visible to infrared translation

Previous works have explored the translation from VIS to IR [2, 10, 13, 22, 28, 31, 38] or from IR to VIS [3, 15, 37, 40, 42], which differs from image translation based on style transfer. Because IR images depict thermal textures, it is crucial to consider the mapping relationship between the two modal features when designing translation methods. Currently, many methods [14, 35, 44] rely on paired multimodal data for training to capture the thermal region features. However, this limits their generalization ability beyond the training dataset, making it challenging to achieve balanced results across different data domains, such as varying infrared wavelengths and imaging backgrounds. Additionally, some studies [19, 34] propose integrating thermal information into the model, utilizing segmentation or mask to highlight object features as salient areas of interest for the model. [7] uses infrared temperature information to guide the preservation of VIS details. The challenge with this method lies in accurately segmenting and simulating thermal region, as well as determining the appropriate weight of thermal features in output. Meanwhile, [13] proposes edge-guided method for multi-domain translation, emphasizing the retention of edge information and crucial details in generated IR images, but this focus sometimes leads to the neglect of structural features in vis images.

### 2.2 Image to image translation

Recently, image-to-image translation methods have rapid development. The majority of translation methods [5, 11, 16, 24, 29, 36, 45, 47] primarily focus on style transfer. Typically, they map images to latent spaces, perform specific conversions or comparisons, and then reconstruct into images. CUT [23] preserves the structural features of the source domain by employing patch contrast loss, whereas CycleGAN [47] incorporates cyclic consistency loss to ensure the structural coherence of the generated image. [4] explores the relationship between global features and instances through enforcing cross-granularity consistency. Nevertheless, convolution operations suffer from a limited receptive range, often resulting in blurring during high-resolution image translation. [24] utilizes autocorrelation regularization to facilitate zero-shot image translation based on diffusion models, but its generation domain is uncontrollable. Recent researches have focused on comprehending image features from the frequency and apply

to image translation. [43] extracts the high-frequency band of images through discrete wavelet transform during knowledge distillation. [46] involves jointly learning the image features in both pixel and Fourier spectral spaces. [16, 33] utilize the Laplace pyramid to segregate images into high and low frequency components, translating contours in low-frequency and enhancing details with high-frequency. Although these methods have generated diversity, the feature differences in multimodal pose challenges for its direct application in VIS-to-IR translation.

## 3 THE PROPOSED METHOD

### 3.1 Network architecture

Given a VIS domain dataset $\{V\} \in \mathbb{R}^{H \times W \times 3}$ and an IR domain dataset $\{I\} \in \mathbb{R}^{H \times W \times 1}$ under similar viewing angles, including VIS images $v \in V$ and IR images $i \in I$. Our goal is to learn the feature mapping that transforms the VIS domain into IR domain, i.e. $G : V \rightarrow I$, and then produced the output image $\widetilde{i}$ which adheres to the infrared thermal laws based on VIS image $v$. The translation process is denoted as follows:

$$\widetilde{i} = f_{V \rightarrow I}(v, \ G(V; I), \ D) \tag{1}$$

As shown in Figure 2, out proposed MappingFormer has two generators $G_V$ and $G_I$, and two discriminators $D_V$ and $D_I$. The generator $G_V$ learn the mapping from VIS to IR, whereas the discriminators $D_V$ assess authenticity. Upon input an image $v \in \mathbb{R}^{H \times W \times 3}$ into the generator, the initial step is a Laplace decomposition to derive low-frequency $v_l \in \mathbb{R}^{\frac{H}{2} \times \frac{W}{2} \times C}$ and high-frequency components $v_h \in \mathbb{R}^{H \times W \times C}$. Then, both components are processed through the encoder-decoder, which embedded with the Swin-T blocks, getting a preliminary IR image generated. Meanwhile, identical steps are applied to real IR images $i \in \mathbb{R}^{H \times W \times 1}$ to achieve corresponding outcomes $\widetilde{v} \in \mathbb{R}^{H \times W \times 3}$. Perform feature patches encoding and space embedding on the dual mapping, followed by computing the corresponding mapping loss. Through network training, domain-transferred IR images $\widetilde{i}$ are attained. Notably, while the training process need the involvement of both image modals, the inference relying solely on the generator $G_V$ to translation the VIS to IR.

### 3.2 Feature mapping

This section describes the feature mapping process within the generator. Inspired from [16, 33], we use the Laplace image pyramid to decompose VIS image into low and high frequency. Then, the Swin-T blocks and encoder-decoder are used to extract and map the feature in two frequency components. Finally, the cross-attention in Swin-T is utilized to fusion the extracted features, resulting in generated images that more accurately reflect the real IR features, as shown in Figure 3. **Low-frequency feature mapping.** The low-frequency component contains visual attributes like illumination and color within the VIS domain. During the translation to the IR domain, illuminations and color features are converted to grayscale values with varying luminosities, as well as the

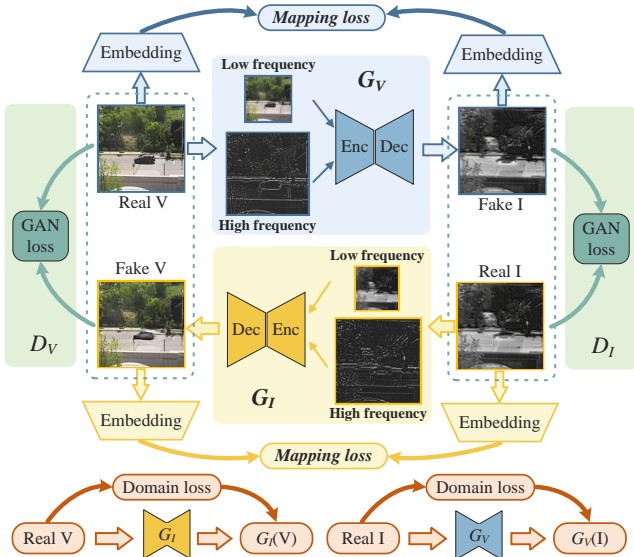

**Figure 2: Overall architecture of MappingFormer. A dual-branch is designed in the generator, while a bidirectional mapping constraint mechanism is introduced. More details are illustrated in section 3.**

consistency of fuzzy structures. Given the low-frequency component $v_l \in \mathbb{R}^{\frac{H}{2} \times \frac{W}{2} \times C}$ of VIS, first conduct feature extraction and channel expansion via encoder structure to acquire the desired feature map $v_l^{(a)} \in \mathbb{R}^{\frac{H}{2^a} \times \frac{W}{2^a} \times 2^{a-1}C}$, $a = [1, 2, 3]$. Then position encoding is added to reserve space information and express the positional relationships of features. Then, six Swin-T blocks are incorporated to efficiently extract the long-distance dependencies from the feature maps. After that, the feature maps are progressively decoded and concatenated, and the low-frequency components are reinstated to the feature maps with original resolution. Both the encoder and decoder are combining convolution with Leaky ReLU to adjust the feature channels, and the resolution of feature maps are adjusted by setting the step size.

The low-frequency mapping branch based on Swin-T blocks efficiently integrates self-attention to the encoder-decoder. This enables the network to accurately capture more prominent feature regions within the low-frequency components. Specifically, the process for calculating low-frequency features in six Swin-T blocks is defined as follows:

$$\begin{aligned}
\hat{x}_l^m &= W\text{-}MSA(LN(x_l^{m-1})) + x_l^{m-1}, \\
x_l^m &= MLP(LN(\hat{x}_l^m)) + \hat{x}_l^m, \\
\hat{x}_l^{m+1} &= SW\text{-}MSA(LN(x_l^m)) + x_l^m, \\
x_l^{m+1} &= MLP(LN(\hat{x}_l^{m+1})) + \hat{x}_l^{m+1},
\end{aligned} \tag{2}$$

where (S)W-MSA and LN represent window self-attention computation and normalization layers, MLP represents multi-layer perceptron, $\hat{x}_l^m$ and $x_l^m$ are the process variables of feature computation, representing the output of (S)W-MSA and MLP for the m-th block structure. Similar to previous

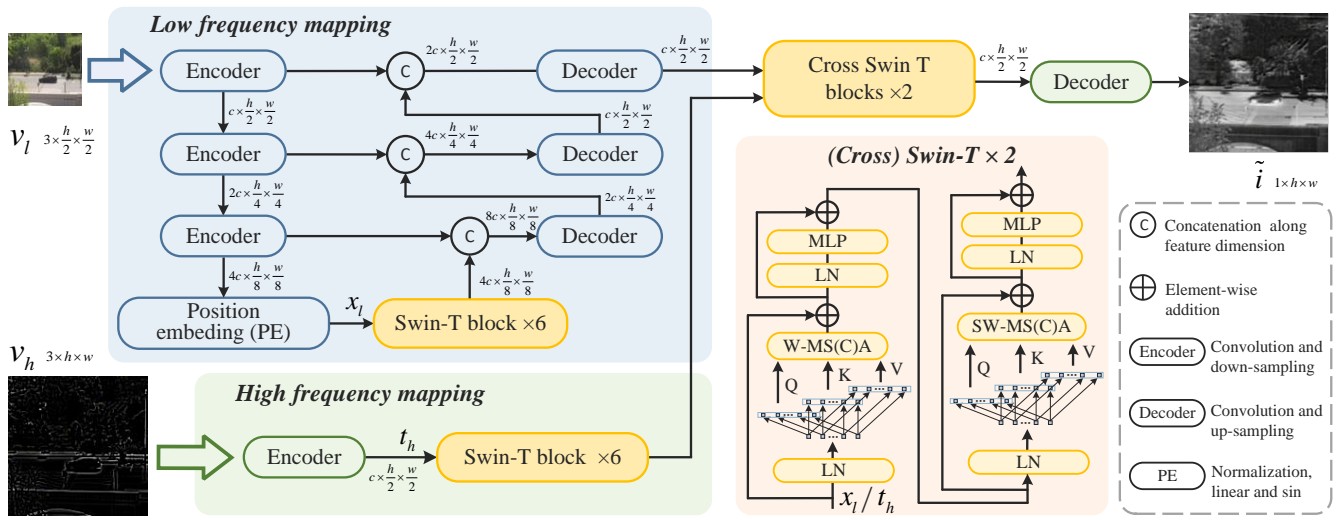

**Figure 3: The two-branch of feature mapping in generator. The two branches are designed: LFM and HFR, both embedded with the Swin-T, to extract and enrich structure and texture details.**

work [20], self-attention calculation is as follows:

$$Attention(Q, K, V) = SoftMax(\frac{QK^T}{\sqrt{d}} + B)V, \quad (3)$$

where $Q$, $K$, $V$, and $d$ represent the query, key, value matrices, and their respective dimensions within the window. $B$ denotes the bias matrix. Furthermore, the low-frequency mapping branch has similarities to the network structure of U-Net [26], where the encoder performs down-sampling, and the decoder handles up-sampling. Shallow-layer encoding captures more detailed visual features, whereas deep-layer encoding extracts richer local information. By concatenating these layers, efficient feature mapping is achieved in the low-frequency domain.

**High-frequency feature refinement.** To reconstruct detailed information in the generated IR domain, such as texture and edges, we use high-frequency component for refinement and supplement. Given the component $v_h \in \mathbb{R}^{H \times W \times C}$ of VIS domain, an encoder is used to perform down-sample, aligning its resolution with that of the low-frequency component. Subsequently, in line with the combined method of the Swin-T blocks described in the low-frequency, self-attention is calculated to identify salient regions within the high-frequency component. Following this, we perform cross-attention calculation based on the cross Swin-T for dual-branch features at a specific resolution $\frac{H}{2} \times \frac{W}{2} \times C$. In this process, the $K$ and $V$ values within the window originate from the low-frequency branch, while $Q$ originates from the high-frequency branch, facilitating the fusion and refinement of feature sequences from different frequency domains. Finally, the output of cross-attention undergoes decoding for up-sampling, resulting in the generation of a preliminary mapping image $\tilde{i}$ . The fusion of features can be considered as supplementing high-frequency features in low-frequency, thereby forming refined features

and producing high-resolution translated images. The calculation of high and low frequency within the cross Swin-T can be described as follows:

$$\begin{aligned}
\hat{t}_h^n &= W \text{-} MCA(LN(t_l^{n-1}), LN(t_h^{n-1})) + t_h^{n-1}, \\
t_h^n &= MLP(LN(\hat{t}_h^n)) + \hat{t}_h^n, \\
\hat{t}_h^{n+1} &= SW \text{-} MCA(LN(t_l^{n-1}), LN(t_h^n)) + t_h^n, \\
t_h^{n+1} &= MLP(LN(\hat{t}_h^{n+1})) + \hat{t}_h^{n+1},
\end{aligned} \quad (4)$$

where (S)W-MCA represents crossing self-attention in window operation, $\hat{t}_h^n$ and $t_h^n$ are the output of (S)W-MCA and MLP in n-th block structure.

### 3.3 Patch-based dual contrastive learning

Based on the CUT[23], we introduce a dual contrastive learning mechanism based on feature patch. This serves as feedback and constraints on the generator's mapping process. The patch-based dual contrastive structure is shown in Figure 4. We simultaneously apply this structure to bidirectional mapping, functioning as a dual contrast in MappingFormer. If a feature block is selected in the generated IR image, the module scouts for comparisons among multiple feature patches derived from the VIS input. It matches and associates the corresponding patches in the VIS image. Specifically, we initially use the patch embedding module to extract feature patches from both the input $v$ and output $\tilde{i}$ of generator $G_V$, obtaining a stack of two feature patches. The patch embedding module consists of a four-layers convolutional encoder and a MLP projection head. Subsequently, when a query is chosen from the IR feature stack, the corresponding feature patches in the VIS are designated as positively correlated, whereas the non-corresponding ones are marked as negatively correlated. We map the query, positive, and negative samples onto K-dimensional vectors $s$, $s^+ \in \mathbb{R}^K$, $s^- \in \mathbb{R}^{N \times K}$ respectively. To prevent spatial expansion or collapse, we normalize

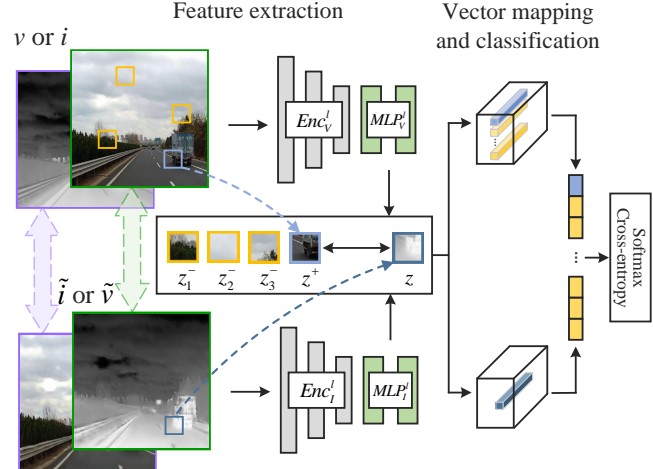

**Figure 4: Patch-based Contrastive Learning. We input two sets of data to simultaneously conduct a bidirectional comparison mechanism.**

these vectors using L2 normalization, confining them to the unit sphere. Following this, we establish a $(N+1)$ directional classification problem and calculate the probability of positive samples surpassing negative ones. The mathematical definition is as follows:

$$\ell(s, s^+, s^-) = -\log\left(\frac{\exp(s \cdot s^+/\tau)}{\exp(s \cdot s^+/\tau) + \sum_{n-1}^{N} \exp(s \cdot s^-/\tau)}\right) \quad (5)$$

where $\tau$ is the scaling distance parameter and is set to a default of 0.07.

## 3.4 Loss function

The total loss of our method comprises the mapping contrastive loss $\mathcal{L}_{mapping}$, adversarial loss $\mathcal{L}_{adv}$, and domain conversion loss $\mathcal{L}_{dom}$.

**Mapping contrastive loss.** We design a patch embedding-based dual contrastive mechanism to perform patch encoding and spatial embedding for the mapping $G_V : V \to I$ and $G_I : I \to V$. By selecting $L$ layers feature patches from the encoder and inputting them into a two-layer MLP projection head, we obtain a feature stack $\{z_l\}_L = \{MLP_V^l(Enc_V^l(v))\}_L$, where $Enc^l$ is the encoded features of $l$-th layer. At this stage, every layer of features within the stack actually represents an image patch. The spatial positions of each layer can be denoted as $p \in \{1, \ldots, P_l\}$, where $P_l$ is the number of spatial positions in each layer. Then we select a query, the corresponding positive sample patch is denoted as $z_l^p \in \mathbb{R}^{C_l}$, and the remaining negative ones are denoted as $z_l^{P \backslash p} \in \mathbb{R}^{(P_l-1) \times C_l}$, where $C_l$ is the number of channels in each feature patch. Similarly, another feature stack $\{\hat{z}_l\}_L = \{MLP_I^l(Enc_I^l(G_V(x)))\}_L$ can be acquired.

In order to match the positive sample patches between input and output, and evaluate the mapping effect of the generator, this section refers to the multi-layer contrast loss based on patch matching [23] as the mapping contrastive loss

$\mathcal{L}_{mapping\_V}$ from visible domain $V$ to infrared domain $I$:

$$\mathcal{L}_{mapping\_V}(G_V, MLP_V, MLP_I, V) =$$
$$\mathbb{E}_{v \sim V} \sum_{l=1}^{L} \sum_{p=1}^{P_l} \ell(\hat{z}_l^p, z_l^p, z_l^{P \backslash p}). \quad (6)$$

For reverse mapping, introduce a similar mapping contrastive loss $\mathcal{L}_{mapping\_I}$ :

$$\mathcal{L}_{mapping\_I}(G_I, MLP_V, MLP_I, I) =$$
$$\mathbb{E}_{i \sim I} \sum_{l=1}^{L} \sum_{p=1}^{P_l} \ell(\hat{z}_l^p, z_l^p, z_l^{P \backslash p}). \quad (7)$$

where the feature stacks $\{z_l\}_L = \{MLP_I^l(Enc_I^l(i))\}_L$, and $\{\hat{z}_l\}_L = \{MLP_V^l(Enc_V^l(G_V(i)))\}_L$. The total mapping contrastive loss $\mathcal{L}_{mapping}$ is the summation of both directions.

**Adversarial loss.** The adversarial loss $\mathcal{L}_{adv}^v$ constrains the similarity between the output image of generator and the real image through discriminator $D_V$ and $D_I$. Based on the dual-branch discriminator introduced in [16], we calculate the losses incurred by the high-frequency and low-frequency branches individually.

**Domain conversion loss.** The generator $G_V$ converts the VIS image into an IR domain image. When sending an IR image to $G_V$, the expected output remains within the IR domain and $G_V(i)$ should closely resemble the original IR image $i \in I$. Meanwhile, the output $G_I(v)$ should also be similar to VIS images $v \in V$. In short, it ensures that the color style of the generated modality is consistent with the real domain. The design of the domain conversion loss is as follows:

$$\mathcal{L}_{dom} = \mathbb{E}_{v \sim V}\left[\|G_I(v) - v\|_1\right] + \mathbb{E}_{i \sim I}\left[\|G_V(i) - i\|_1\right] \quad (8)$$

**Total loss.** The total loss $\mathcal{L}$ is calculated using the following combination:

$$\mathcal{L} = \lambda_1 \cdot \mathcal{L}_{mapping} + \lambda_2 \cdot \mathcal{L}_{adv} + \lambda_3 \cdot \mathcal{L}_{dom} \quad (9)$$

where $\lambda_1$, $\lambda_2$, and $\lambda_3$ are coefficients that respectively adjust the weight of mapping contrastive loss, adversarial loss, and domain conversion loss.

## 4 EXPERIMENTS

### 4.1 Datasets and details

**AVIID-1 dataset [6]:** This public dataset focuses on vehicles in road scene, including objects such as cars, buses, small trucks, and larger trucks. It includes 993 pairs of VIS and IR images, each with the resolution of $434 \times 434$.

**IRVI dataset [15]:** This public dataset shows vehicle driving scenes, mainly capturing vehicles and backgrounds from a forward perspective while driving on the road. The dataset comprises 17,000 pairs of training images and 1,000 pairs of test images, each with a resolution of $256 \times 256$.

**The DroneCoast dataset:** Currently, there are limited publicly datasets for aerial coast scenes captured using both visible and infrared. Hence, our work employs binocular mid-wave infrared and color cameras to record coast scene video streams, thereby constructing an unaligned dataset. This

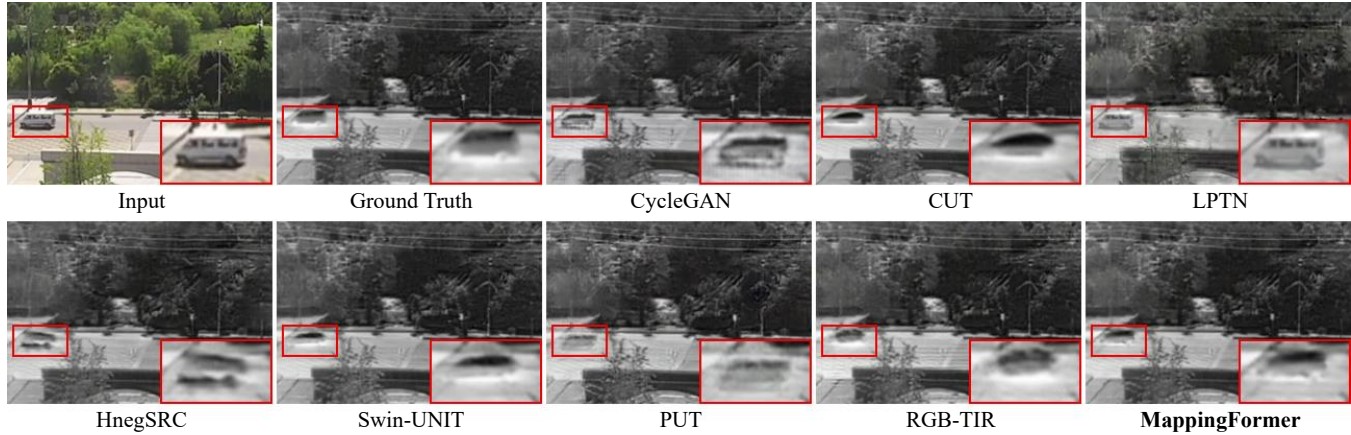

**Figure 5: Qualitative comparisons on the AVIID-1 dataset (Monitor scene).**

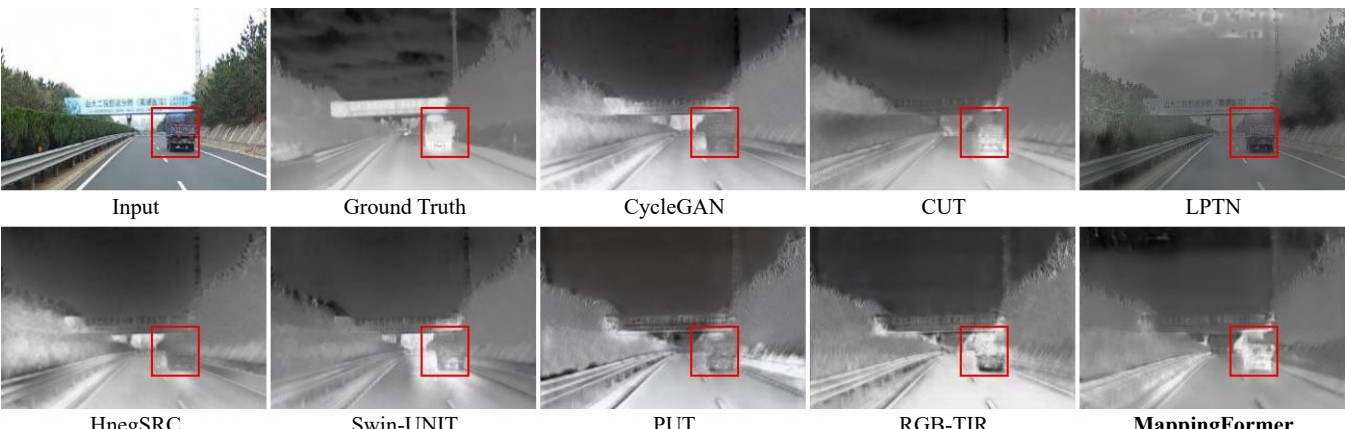

**Figure 6: Qualitative comparisons on the IRVI dataset (Driving scene).**

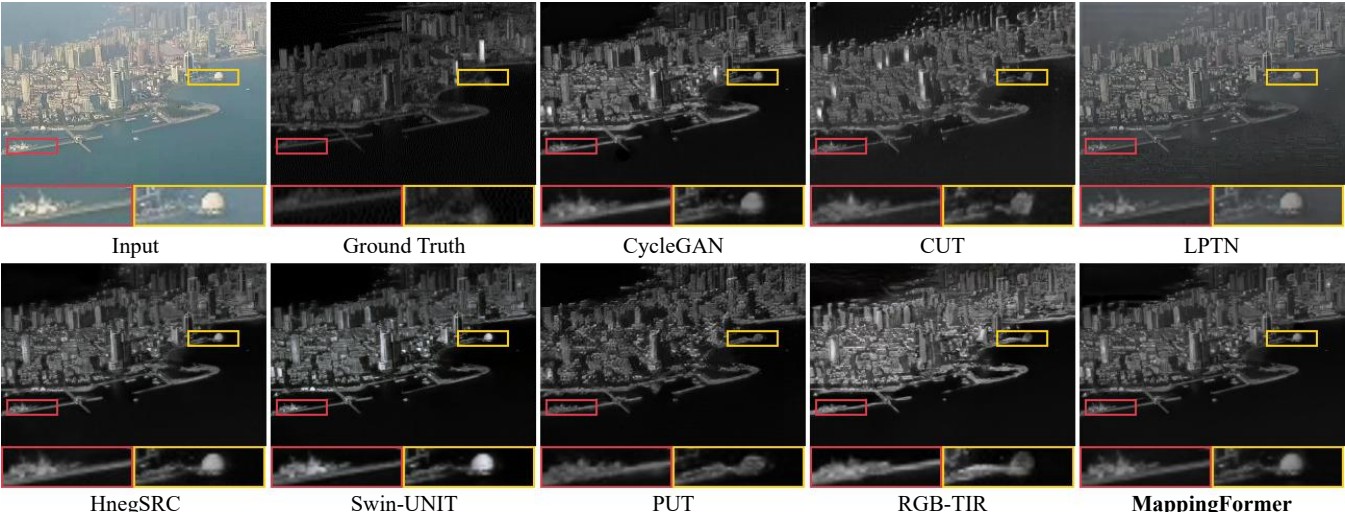

**Figure 7: Qualitative comparisons on the DroneCoast dataset (Coast scene).**

**Table 1: Quantitative comparison on three datasets. Bold and underline denote the best and suboptimal performance, respectively.**

| Public | Method | AVIID-1 | | | IRVI | | | DroneCoast | | |
|--------|--------|---------|---------|------|---------|---------|------|---------|---------|------|
| | | SSIM↑ | PSNR↑ | FID↓ | SSIM↑ | PSNR↑ | FID↓ | SSIM↑ | PSNR↑ | FID↓ |
| ICCV 17 [47] | CycleGAN | 0.71 | 24.61 | 0.59 | 0.56 | 18.52 | 1.11 | 0.68 | 24.24 | 1.71 |
| ECCV 20 [23] | CUT | 0.72 | 24.45 | 0.69 | 0.63 | **20.77** | 0.90 | 0.63 | 22.90 | 1.41 |
| CVPR 21 [17] | LPTN | 0.73 | 20.52 | 1.86 | 0.53 | 17.49 | 1.32 | 0.61 | 17.60 | 2.61 |
| MM 23 [16] | Swin-UNIT | 0.80 | 25.96 | 0.87 | 0.65 | 18.92 | 0.89 | 0.73 | 25.16 | 1.28 |
| MM 22 [18] | PUT | 0.70 | 24.14 | 0.38 | 0.63 | 19.31 | 1.15 | 0.64 | 23.49 | 1.39 |
| CVPR 22 [11] | HnegSRC | 0.59 | 22.43 | 0.54 | 0.57 | 18.57 | 1.31 | 0.70 | 22.85 | 1.54 |
| ICRA 23 [13] | RGB-TIR | 0.81 | 25.59 | **0.37** | 0.62 | 19.98 | 1.34 | 0.71 | 24.72 | 1.76 |
| Ours | MappingFormer | **0.86** | **27.52** | 0.55 | **0.68** | 20.38 | **0.70** | **0.79** | **25.48** | **1.01** |

dataset contains 939 pairs of training images and 164 pairs of test images, each with a resolution of $640 \times 512$.

**Implementation details:** The batch size in training is set to 1, with an initial learning rate set to $1 \times 10^{-4}$. The training epochs are 200, with a 50 % decay every 50 epochs for learning rate, and set the loss weights $\lambda_1 = 2$, $\lambda_2 = 1$, $\lambda_3 = 1$. All experiments were conducted on a workstation with a 4090 GPU. We implemented and compared our method with advanced general image translation and VIS-to-IR translation methods, including CycleGAN [47], CUT [23], LPTN [17], HnegSRC [11], Swin-UNIT [16], PUT [5], and RGB-TIR [13].

## 4.2 Evaluation metrics

Structural similarity (SSIM) is a metric to assess the likeness between generated and real images. The value closer to 1, the stronger resemblance between the generated and the real image. The peak signal-to-noise ratio (PSNR) quantifies the distortion between the generated and the real image. A higher PSNR indicates to lesser distortion in the generated image. The Frechet Inception Distance (FID) compares the feature distributions of real and generated images by computing depth features extracted using the Inception-V3 model. A lower FID value implies a closer distribution between the two images in the feature space. This paper normalized all the FID metrics. Furthermore, Section 4.6 assesses the performance of IR images translated by MappingFormer in downstream tasks, using precision, recall, and average precision (AP) under various intersection-over-union in object detection.

## 4.3 Visual qualitative comparison

The visualization of test results on three datasets is shown in Figure 5, Figure 6, and Figure 7. Evidently, CUT performs well on the low-resolution traffic scenes in Figure 5 and Figure 6. However, it causes ghosting shadow in the high-resolution scene shown in Figure 7. We consider it might stem from inadequate constraints by the only one generator. Although CycleGAN and HnegSRC are capable of converting images from the VIS domain to IR, they often retain an excess of VIS features, thereby not adhering to the IR characteristics. PUT might introduce blurring, especially in Figure 6 and

Figure 7. Despite utilizing Laplace's high and low frequency components, LPTN still maintains the original color style and fails to translate images to the IR domain. RGB-TIR accurately highlights the heating area of the truck in Figure 6, but its overall brightness is elevated and the generated texture features lack smoothness. Swin-UNIT's performance is suboptimal, presumably due to its two-stage approach involving global style translation and recurrent detail supplementation, which nonetheless falls short in capturing the essential thermal features of IR images. The visual results evident that our proposed method is superior in generating realistic IR thermal features while maximally preserving the intricate details in VIS images.

## 4.4 Quantitative evaluation

Table 1 shows the quantitative evaluation of three datasets for VIS-to-IR translation. As evident from the table, our method achieved the best result in 7 indicators and achieved suboptimal in 1 indicator. Specifically, MappingFormer's SSIM metric on the AVIID-1, IRVI, and DroneCoast datasets were 0.86, 0.68 and 0.79, respectively. These results outperformed other advanced general image translation methods and VIS-to-IR translation methods, exceeding the suboptimal by 5%, 3% and 6%, respectively. The results strongly suggest that MappingFormer's frequency feature mapping, coupled with dual contrast learning, is suitable for VIS-to-IR cross-modal translation. Furthermore, Swin-UNIT shows promising performance, achieving suboptimal results in six metrics, which also proved the potential of applying frequency components to image generation.

## 4.5 Ablation analysis

**Effect of the network structure.** We analyze the effect of various network components to the overall performance on AVIID-1 dataset. The results are shown in Table 2. Initially, the baseline model, CUT, was trained using same settings. Afterward, we incrementally introduced LFM, HFR and DFPC to assess how these structures influence performance. Note that both the LFM and HFR branches incorporate Swin-T blocks. Model-1 and Model-2 integrated the LFM and HFR

**Table 2: Key component analysis in AVIID-1 dataset**

| Model | LFM | HFR | DFPC | SSIM↑ | FID↓ |
|---|---|---|---|---|---|
| Baseline | | | | 0.717 | 0.686 |
| Model-1 | ✓ | | | 0.762 | 0.657 |
| Model-2 | | ✓ | | 0.775 | 0.649 |
| Model-3 | ✓ | ✓ | | 0.822 | 0.583 |
| MappingFormer | ✓ | ✓ | ✓ | **0.858** | **0.547** |

**Table 3: Different Swin-T blocks analysis in AVIID-1**

| Model | Generator | SSIM↑ | FID↓ |
|---|---|---|---|
| Model-1 | Conv-based | 0.761 | 0.662 |
| Model-2 | Swin-T in LFM | 0.796 | 0.593 |
| Model-3 | Swin-T in HFR | 0.803 | 0.620 |
| Model-4 | Swin-T in LFM/HFR | 0.831 | 0.577 |
| Model-5 | Model-4 + Cross Swin-T | **0.858** | **0.547** |

branches, respectively, and the SSIM metric improved 4.5% and 5.8% compared to the baseline. Model-3 incorporates both LFM and HFR branches, yielding a substantial 10.5% SSIM improvement over the baseline. When all three modules are used concurrently, both SSIM and FID metrics exhibit a noteworthy enhancement. In conclusion, MappingFormer has achieved leading VIS-to-IR image translation effects. The design of the feature mapping branch is crucial, while the dual patch comparison also contributes meaningfully to the overall performance.

**Effect of the Swin transformer block.** We explore the effect of embedding Swin-T blocks within feature mapping on AVIID-1 dataset. We incorporate block structures into one mapping branch while maintaining convolution structures in the other, and then contrast them with models that only rely on convolutional designs. Our findings indicate that embedding Swin-T blocks within the mapping branch yields higher quality images when compared to generators that are based solely on convolution structures. Note that each model fuses the outputs of high and low frequency branches, but only Model-5 uses cross Swin-T blocks during the fusion process, whereas other models rely solely on an accumulator for fusion. The detailed results are shown in Table 3.

## 4.6 Extension to object detection

This section aims to apply the outcomes generated by proposed MappingFormer to downstream object detection. For a quantitative evaluation, we trained and tested both our method and baseline network on the AVIID-1 dataset, obtaining images translated from the VIS domain to IR domain. We chose the latest Yolov9 [32] as our benchmark for object detection. To prevent data leakage, the IR images generated by two translation models serve as the training set for object detection respectively, while the unused real IR images constitute the test set. Detection results are calculated using the above training data. Additionally, we compare the detection

**Table 4: Different training data for object detection**

| Data source | Precision | Recall | AP_50 | AP_50:95 |
|---|---|---|---|---|
| MappingFormer | 0.950 | 0.872 | 0.937 | 0.518 |
| CUT | 0.917 | 0.806 | 0.879 | 0.460 |
| Trained in real visible | 0.676 | 0.462 | 0.544 | 0.208 |
| Trained in real infrared | **0.964** | **0.896** | **0.952** | **0.564** |

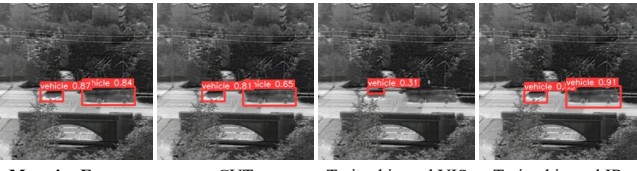

| MappingFormer | CUT | Trained in real VIS | Trained in real IR |

**Figure 8: Applying training data from different sources to object detection.**

performance of training directly on real VIS or real IR images. The Yolo test results are shown in Table 4, and Figure 8 shows the detection visualization.

Based on detection performance, directly training with VIS images and performing cross-domain detection yields poor results, lacking of generalization ability between the multimodal. Using IR images generated by MappingFormer as training data for detection achieves comparable performance compared to training directly with real IR data. Although our detection results do not surpass those trained on real IR images, this is reasonable because generative models can only simulate the distribution of real data to the best of their ability. Experiment results indicate that the feature distribution of the IR images generated by proposed method closely resembles that of real IR, making them effectively applicable to downstream tasks.

## 5 CONCLUSION

In this paper, we propose a novel approach for image translation from VIS to IR. We have designed a specialized framework, MappingFormer, which integrates feature mapping and contrastive learning within a generator network for cross-modal image translation. Within the generator, two branches are designed: low-frequency feature mapping and high-frequency feature refinement, both of which are embedded with the Swin Transformer. Additionally, the dual contrastive structure based on feature patches serves as a constraint for mapping and generation. Experiment results indicate that the proposed method surpasses general image-to-image generation methods in both qualitative and quantitative evaluations, and outperforms the advanced VIS-to-IR translation methods. These experiments confirm the efficacy of feature mapping and dual contrastive learning. Future work could explore the temporal of video translation and unsupervised generation of multimodal images within VIS and IR domain.

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
