# OpenReview forum: "MappingFormer: Learning cross-modal feature mapping for visible-to-infrared image translation"
_acmmm.org/ACMMM/2024/Conference — MM2024 Poster_

### Official Review · Reviewer_qgjb · 2024-05-09

**Rating:** 3
**Confidence:** 4

**Summary:**

This article mainly introduces a cross-modal feature mapping method for visible images to infrared image conversion, namely MappingFormer, which achieves feature mapping by mining frequency features between image domains.The Swin Transformer is used to process high-frequency and low-frequency features separately, and the extracted dual-branch frequency features are refined and fused through a cross-attention mechanism. A dual-contrast learning mechanism (DFPC) based on feature blocks is designed to infer effective mappings between unaligned cross-modal data to improve the mapping effect.

**Strengths:**

1. This article is clearly illustrated and written, making it easy for readers to understand the main ideas and structure.
2. The experimental results of this paper are sufficient to provide strong experimental evidence for the effectiveness of MappingFormer in implementing VI to IR.
3. Under the condition of limited infrared detectors, this paper proposes an effective remedy with certain practical value.

**Limitations:**

1. This paper is just a combination of several existing proposed algorithm modules, with very few new proposed works. It lacks novelty and true ideological innovation.In my humble opinion,the main idea of the article seems to be a combination of LPTN and ROMA, with only the novel idea of decoupling and merging frequency-domain features using Swin-T. However, this single idea is not sufficient to justify the value of the entire article.
2. The article's design explanation for some structures is not complete, such as why only one encoder is used for high-frequency features. Although the shallow features have better original structures, how many layers are used is obtained through experimental testing or defined independently based on previous work experience. Please provide an appropriate explanation.

**Suitability:**

2

---

### Official Review · Reviewer_nyd6 · 2024-05-18

**Rating:** 5
**Confidence:** 4

**Summary:**

The paper addresses the challenge of limited infrared data availability by proposing a novel approach called MappingFormer for cross-modal image generation from visible to infrared. Current methods often fail to consider the crucial infrared thermal features during the conversion process, leading to suboptimal performance. MappingFormer leverages frequency feature mapping and dual patches contrast to enhance the authenticity of cross-modal translation at the feature level. The model consists of a generator with two branches - low-frequency feature mapping (LFM) and high-frequency feature refinement (HFR) - both utilizing Swin Transformer blocks. The LFM branch captures structural features from visible images, while the HFR branch focuses on edge and texture features. The refined features are fused through cross-attention mechanisms, and a dual contrast learning mechanism based on feature patches (DFPC) further enhances the alignment between unaligned cross-modal data. Experimental results demonstrate the effectiveness of MappingFormer in cross-modal feature mapping and image generation, showing promise for downstream tasks with limited infrared data availability.

**Strengths:**

1. MappingFormer excels in preserving the authenticity of cross-modal translation at the feature level by incorporating frequency feature mapping, dual patches contrast, and Swin Transformer blocks. This enhances the accuracy and quality of generated infrared images from visible inputs.
2. The dual-branch generator design, with LFM and HFR branches focusing on different aspects of feature extraction, allows for a comprehensive representation of visible images that facilitates accurate mapping to infrared images. The fusion of these features through cross-attention mechanisms enhances the translation process.
3. The introduction of the dual contrast learning mechanism based on feature patches (DFPC) addresses the challenge of aligning unaligned cross-modal data, improving the effectiveness of feature mappings between visible and infrared images.
4. The paper provides comprehensive experimental results that demonstrate the effectiveness of MappingFormer in cross-modal feature mapping and image generation. This empirical evidence supports the potential of the proposed method for tasks requiring the translation of visible images to infrared in scenarios where infrared data is scarce.

**Limitations:**

1. The practical applicability and scalability of MappingFormer in real-world scenarios with diverse infrared image acquisition conditions are not discussed.
2. While the proposed model architecture appears to be innovative and effective, the abstract does not address the computational complexity or resource requirements associated with implementing MappingFormer.

**Suitability:**

3

---

### Official Review · Reviewer_EevG · 2024-05-25

**Rating:** 4
**Confidence:** 2

**Summary:**

The authors introduce a translation network based on frequency feature mapping and dual patches contrast, Mapping Former, which can achieve cross-modal image generation from visible to infrared to alleviate the critical shortage of infrared data.  In this paper, a cross-modal feature mapping network is proposed for VIS-to-IR (MappingFormer) task. An important new idea is proposed, and its effectiveness is verified by experiments. However, further detailed comparative experiments, description of technical details and discussion of practical applications will enhance the integrity of the paper.

**Strengths:**

1. MappingFormer improves the authenticity of cross-modal image translation through frequency feature mapping and two-patch contrast learning mechanism, which is a novel attempt in the field of visible light to infrared image conversion.

2. In the design of the network structure, they design a two-branch generator that includes low frequency feature mapping (LFM) and high frequency feature thinning (HFR). This structure helps to process different features of images separately and improve the translation quality. Swin Transformer blocks are embedded in the LFM and HFR branches, which helps to efficiently extract long distance dependent features and enhance model perception.

3. By contrast learning mechanism, the bidirectional contrast learning mechanism (DFPC) based on feature patches was designed to better learn the effective mapping between unaligned cross-modal data.

**Limitations:**

1. In the contribution part, it is mentioned that it is the first work that applies frequency mapping to visible light to infrared conversion, but in the related work part, it is mentioned that recent studies mainly focus on understanding image features from frequency, which is not fully stated.
2.In the process of comparative experiments, experimental data show that the model has certain competitiveness, but it is not particularly outstanding. Can you further explain the performance of the model in terms of other evaluation indicators such as the number of parameters?
3.From the perspective of model complexity, the introduced double-branch structure and contrast learning mechanism may increase the complexity of the model, which may affect the training and reasoning efficiency of the model. Please add the explanation in this part.
4.In Figure 3, the W-MSA and W-MCA mentioned in this paper are not well shown. This is not intuitive enough for the reader. Please mark them in Figure 3.
5. In Extension to object detection section, only one method, CUT, is used for comparison. Vis-to-IR methods like Swin-UNIT, PUT, and RGB-TIR, are not mentioned in this experiment.
6. In Equation 5, n is never mentioned in this section, which is ambiguous to the reader. Please provide an explanation of the meaning of this symbol after the introduction of the equation.

**Suitability:**

3

---

### Official Review · Reviewer_zJYS · 2024-05-25

**Rating:** 3
**Confidence:** 3

**Summary:**

This paper presents an approach for image translation from VIS to IR. It designed a MappingFormer to integrate feature mapping and contrastive learning within a generator network for crossmodal image translation. Within the generator, two branches are designed: low-frequency feature mapping and high frequency feature refinement, both of which are embedded with the Swin Transformer. Additionally, the dual contrastive structure based on feature patches serves as a constraint for mapping and generation.

**Strengths:**

Experiment results indicate that the proposed method surpasses general image-to-image generation methods in both qualitative and quantitative evaluations. The paper also shows the extension results to object detection.

**Limitations:**

1. How generalizable is the proposed VIS to IR conversion method? For example, in real-world scenarios, across datasets, etc.
2. What is the computational complexity and the number of parameters of the proposed method? It should be compared with existing methods.
3. I am not quite clear about what challenges in the image conversion task the proposed method addresses, as there are many cyclic conversion methods. What are its advantages compared to existing methods?
4. Can you explain the effectiveness of the proposed VIS to IR conversion method mathematically?

**Suitability:**

2

---

### Meta-Review · Area_Chair_Re7A · 2024-06-30

**Recommendation:** Accept (Poster)
**Confidence:** 5

**Metareview:**

This paper initially got 2 accepts and 2 rejects, but all reviewers raised the scores after rebuttal. The authors have addressed the concerns from the reviewer.